# Mg-Doped ZnO Nanoparticles with Tunable Band Gaps for Surface-Enhanced Raman Scattering (SERS)-Based Sensing

**DOI:** 10.3390/nano12203564

**Published:** 2022-10-12

**Authors:** Samuel Adesoye, Saqer Al Abdullah, Kyle Nowlin, Kristen Dellinger

**Affiliations:** 1Department of Nanoengineering, Joint School of Nanoscience and Nanoengineering, North Carolina A&T State University, 2907 E Gate City Blvd, Greensboro, NC 27401, USA; 2Department of Nanoscience, Joint School of Nanoscience and Nanoengineering, University of North Carolina at Greensboro, 2907 E Gate City Blvd, Greensboro, NC 27401, USA

**Keywords:** surface-enhanced Raman scattering, enhancement factor, metal oxides, stability, cell viability, cytotoxicity, substitutional doping

## Abstract

Semiconductors have great potential as surface-enhanced Raman scattering (SERS) substrates due to their excellent physiochemical properties. However, they provide low signal enhancements relative to their plasmonic counterparts, which necessitates innovation in their synthesis and application. Substitutional atomic doping is proposed to improve SERS enhancement by controlling electronic properties, such as the band gap. In this work, zinc oxide (ZnO) nanoparticles were synthesized by co-precipitation and doped with magnesium (Mg) at concentrations ranging from 2–10%. Nanoparticle morphology and size were obtained by scanning electron microscopy (SEM). Elemental composition and chemical states were determined using X-ray photoelectron spectroscopy (XPS). Optical properties were obtained with a UV-vis spectrophotometer, while a Raman spectrometer was used to acquire Raman signal enhancements. Stability was assessed by UV-vis spectroscopy, while cytotoxicity was evaluated by the 3-(4,5-dimethylthiazol-2-yl)-2,5-diphenyltetrazolium bromide (MTT) assay. The results showed that the absorption edge of Mg-doped ZnO nanoparticles was red-shifted compared to pure ZnO nanoparticles. The band gap decreased (3.3–3.01 eV) with increasing Mg doping, while the highest Raman enhancement was observed at 2% doping. No significant cytotoxic effects were observed at low concentrations (3–12 μg/mL). Overall, this study provides evidence for the tunability of ZnO substrates and may serve as a platform for applications in molecular biosensing.

## 1. Introduction

The reliable detection of target molecules, including environmental contaminants, diagnostic biomarkers, cells, pathogens, and extracellular vesicles, is a growing and diverse field of research that necessitates highly specific and sensitive techniques. Surface-enhanced Raman scattering (SERS) is one such technique that has been widely used in various detection strategies [1,2,3,4]. SERS is a phenomenon that yields the amplification of Raman signals via interactions with nanostructured substrates and is understood to be a product of electromagnetic and chemical mechanisms. The electromagnetic mechanism is most widely attributed to plasmon excitation in metal nanostructures (e.g., gold or silver nanoparticles) functioning as substrates. The chemical mechanism of enhancement is broadly attributed to a group of processes associated with the transfer of electrons between a molecule and a target substrate [5]. Semiconductors have recently been proposed as SERS substrates due to their excellent physical properties, unique chemistries, and tunable interactions with light [6,7,8]. Leveraging the chemical enhancement mechanism, when a semiconductor is used as a SERS substrate, an electronic transition occurs between the semiconductor and the molecule of interest (e.g., target analyte or Raman reporter), which depends on their respective energy levels and the coupling between them [7,9,10]. The electronic transitions generally considered for this enhancement include the excitonic transition between the valence band (VB) and conduction band (CB) of the substrate, a molecular transition between the highest occupied molecular orbital (HOMO) and lowest unoccupied molecular orbital (LUMO) of the molecule, a charge-transfer transition from the HOMO of the molecule to the CB of the substrate, and a charge-transfer transition from the VB of the substrate to the LUMO of the molecule [9].

Towards developing novel SERS substrates or nanotags for bioanalytical and bioimaging applications, several considerations need to be made, including stability in biological milieus, reliability, and potential toxicity. Zinc oxide (ZnO) is particularly relevant to addressing traditional limitations, with properties that include a wide band gap, high binding energy, large surface-to-volume ratio, and bandgap tunability, with wide availability and low raw material costs [8,11]. In addition, synthesis is flexible and reproducible, and ZnO nanostructures are generally considered stable and biocompatible [12,13,14]. However, ZnO applications in SERS are still limited due to a low enhancement efficiency, particularly compared to traditional SERS substrate materials (e.g., gold and silver).

To improve the enhancement, strategies, such as coating with plasmonic materials to create hybrid SERS substrates [11] and substitutional doping [15], which introduces defects into the lattice to improve the charge-transfer process between ZnO and the target molecule, have been proposed. Of particular note is doping, which results in changes in the lattice constant, bond energy, and energy gap [6]. When an appropriate amount of dopant is introduced to ZnO, surface defect energy levels are formed, and the separation efficiency of the charges is improved, preventing electron–hole recombination [16,17]. In other words, doping increases defects, such as oxygen vacancies, that can induce new surface-state energy levels in ZnO to aid the charge-transfer process. This allows for an easier transition of electrons between various energy levels to release Raman photons [18]. A reduced band gap also results from doping and is speculated to allow for an easier transition of electrons by providing a resonance match between the laser excitation energy and charge-transfer transition [19]. Various dopants, such as Mg [16], Co [20], and Ga [19], have been used to introduce defects into ZnO. For instance, Mg has been used to dope ZnO due to its high solubility in the ZnO lattice [16] and comparable radius to Zn, which could prevent lattice distortion. The bandgap shift resulting from doping ZnO with Mg has been studied in different experiments exploring various applications; while an increased band gap was reported in some experiments [21,22,23,24], others reported a decrease in the band gap [25,26,27]. From this, we can deduce that doping ZnO with different concentrations of Mg can be used as a mechanism to tune the band gap and tailor it for better performance, as demonstrated by research on humidity sensors [27], antibacterial applications [25], and photocatalytic and photodegradation activities [24,26]. For cellular applications, particularly live-cell imaging, acquiring the SERS spectrum of analytes directly from a colloidal substrate is important and could pave the way for SERS in clinical applications.

In this work, Mg-doped ZnO nanoparticles were synthesized by a co-precipitation method to yield structures with bandgap shifts for applications in SERS. For the first time, an enhanced Raman signal effect was observed on molecules using a colloidal substrate of pure and Mg-doped ZnO, which is appropriate for cellular applications and in point-of-care environments where resources are limited. In addition, stability tests and dose-dependent doping for toxicity responses in live cells were carried out to determine the suitability of the substrates in a biological environment. We anticipate that this research will serve as a platform to improve SERS semiconductor-based substrate materials for biomolecular sensing and provide the reliable detection of target molecules, including diagnostic biomarkers, cells, pathogens, and extracellular vesicles.

## 2. Materials and Methods

### 2.1. Chemicals and Reagents

Zinc acetate dihydrate was purchased from Sigma Aldrich (St. Louis, MO, USA); magnesium acetate tetrahydrate was purchased from Fisher Scientific (Waltham, MA, USA); sodium hydroxide was purchased from Sigma Aldrich; and 5,5′-dithiobis (2-nitrobenzoic acid) (DTNB) was purchased from Thermo Fisher Scientific (Heysham, UK). MCF-7 cells, Dulbecco’s Modified Eagle’s Medium (DMEM), penicillin/streptomycin, and fetal bovine serum (FBS) were purchased from ATCC (Manassas, VA, USA). The CyQuant™ MTT (3-(4,5-Dimethylthiazol-2-yl)-2,5-Diphenyltetrazolium Bromide) Cell Viability Assay was purchased from Thermo Fisher Scientific (Waltham, MA, USA).

### 2.2. Preparation of ZnO and Mg-Doped ZnO Nanoparticles

The synthesis of ZnO nanoparticles was carried out using a co-precipitation method [26]. Briefly, for pure ZnO, equal volumes of 0.1 M ZnAce and 0.2 M NaOH were stirred at 750 rpm and heated to 60 °C for 2 h. The product was then centrifuged at 10,000 rpm for 5 min, washed four times with water, and dried for 6 h at 75 °C. For Mg-doped ZnO, Zn_1−x_Mg_x_O (X = 0.02, 0.05, 0.07, 0.1) samples were prepared by adding the calculated amount of 0.1 M Mg(CH_3_COO)_2_∙4H_2_O to NaOH, which was stirred for 1 h. The appropriate amount of ZnAce (based on the desired doping ratio) was added, and the mixture was heated for 2 h. The product was centrifuged at 10,000 rpm, washed four times with water, and dried in the oven, as previously described.

### 2.3. Characterization

The morphology and size of the nanoparticles were obtained using a Zeiss Auriga field-emission scanning electron microscope (FE-SEM) and further analyzed using ImageJ. An ESCALAB™ X-ray photoelectron spectrometer (XPS) was used to determine the elemental composition and the binding energy of the materials. Optical properties were obtained with an Agilent 3000i UV-vis spectrophotometer.

### 2.4. Raman Signal Measurement

The Raman signal was acquired using an i-Raman Prime portable Raman spectrometer (B&W Tek, Newark, DE, USA) equipped with a 785 nm laser source. This spectrometer has an accessory for acquiring the Raman signal of liquid samples in a compatible vial. Thus, the Raman signals (I_Raman_) were acquired directly from the colloidal substrate mixed with a solution of DTNB, the model compound, before acquiring the SERS signals (I_SERS_). In brief, a 10^−2^ M DTNB solution was placed in a glass vial and positioned in the liquid compartment of the Raman spectrometer to acquire the Raman signal, while 0.5 mM of each substrate was dispersed with 10^−2^ M DTNB and sonicated before acquiring the SERS signal. The acquisition time was 1000 ms at 100% laser power and an accumulation number of 3.

### 2.5. Colloidal Stability

Nanoparticle stability in biological mediums is commonly measured by either optical spectroscopy, dynamic light scattering, or microscopy [28]. To monitor the stability of the pure and Mg-doped ZnO substrates, UV-vis spectroscopy was selected because it is well developed and a non-destructive analytical technique [29]. Briefly, stability was evaluated by observing the substrates’ UV-vis absorbance spectra as a function of time in cell culture media [30]. A reduction in this peak indicates a decrease in the concentration of single nanoparticles due to their aggregation over time [28]. In this experiment, the colloidal substrates were dispersed in media, and absorbance was measured with an Agilent 7000i UV-vis spectrophotometer. To prepare the samples, 0.5 mM of each colloidal substrate was added to cell culture media and sonicated for 10 min. Next, the absorbance was measured at 15 min intervals over a time course of 24 h.

### 2.6. Cytotoxicity

The cytotoxicity of these substrates was evaluated by the 3-(4,5-dimethylthiazol-2-yl)-2,5-diphenyltetrazolium bromide (MTT) assay [31,32]. MCF-7 cells were seeded in 96-well plates at 5000 cells per well using Dulbecco’s Modified Eagle’s Medium (DMEM) supplemented with 1% penicillin/streptomycin and 10% fetal bovine serum. Cells were incubated for 24 h at 37 °C in a humidified 5% CO_2_ atmosphere. Thereafter, the cells were treated with several concentrations of pure ZnO, 2% Mg-doped ZnO, 5% Mg-doped ZnO, 7% Mg-doped ZnO, and 10% Mg-doped ZnO. After adding these preparations, cells were incubated for 24, 36, and 48 h. Subsequently, 50 µL of MTT (5 mg/mL) was added to each well prior to incubation for another 3 h. Formazan crystals were dissolved by adding 100 µL of dimethyl sulfoxide (DMSO) to each well, and the absorbance was measured at 540 nm using a multi-well spectrophotometer (B&W Tek, Newark, DE, USA). Untreated cells were used as a baseline control. To analyze data, a two-way analysis of variance (ANOVA) with the Bonferroni test was used for multiple comparisons, and Student’s *t*-test was used to test the significance of differences among the treatment groups. The results are shown as the mean of three independent experiments and the standard error of the mean (±SEM).

## 3. Results and Discussion

### 3.1. Synthesis and Characterization of ZnO and ZnO-Doped Nanoparticles

FE-SEM was used to observe morphological changes resulting from doping the ZnO samples with Mg. From the FE-SEM images (Figure 1), it was observed that pure ZnO nanoparticles had a more uniform size distribution compared to Mg-doped-ZnO nanoparticles. It can also be observed that there is a moderate size increase with an increase in the dopant amount and the agglomeration of smaller particles on the surface, likely due to a higher surface energy [25,26]. In a representative synthesis, pure ZnO had an average size of 75 nm, while the Mg-doped samples had average sizes of 88 nm, 112 nm, 117 nm, and 125 nm for 2%, 5%, 7%, and 10% doping ratios, respectively (see Figure A1 for distribution data). These results likely developed from the successful incorporation of Mg^2+^ into the crystal lattice of ZnO and might enhance the charge-transfer process, which can be favorable for SERS. There is not a universal size or morphology ideal for SERS; however, these results indicate that variations in doping may yield substrates that can be tailored to a desired application.

The elemental composition and relevant chemical states of each sample were obtained from XPS. Figure 2a shows the XPS spectra for the pure and doped ZnO samples. The scans show characteristic peaks of Zn and O in all samples, while the peak of Mg can be seen only in the doped samples. Figure 2b depicts the scan of Zn 2P with peaks around 1020.3 eV and 1043.4 eV, which can be attributed to the binding energy lines of Zn 2p_3/2_ and Zn 2p_1/2_, respectively [26]. The 23.1 eV peak separation indicates that the Zn atom is in the +2 oxidation state [27]. These energy levels are ~0.3 eV and ~0.2 eV lower than the peak positions of the Mg-doped ZnO nanoparticles (1020.6 eV and 1043.6 eV). This is likely due to Mg’s lower electronegativity value compared to Zn, causing Zn 2p to shift when Mg is incorporated into the ZnO lattice [16]. The O 1s peak of the high-resolution spectra of pure and Mg-doped ZnO nanoparticles was fitted into two Gaussian peaks (Figure 2c,d). The lower-energy peak is centered around 529.03 eV and corresponds to 0^2−^ on the wurtzite structure of Zn^2+^ [16,26,27]. In comparison, the higher-energy peak is around 530.39 eV and can be attributed to oxygen vacancies [16]. When Mg was incorporated, this higher-energy peak of O 1s showed a slight shift towards a higher binding energy and higher intensity resulting from increased oxygen vacancy defects [16,27]. The Mg 1s peak located around 1303.5 eV and Mg 2p located around 49.2 eV can be attributed to the magnesium metal, indicating that Mg^2+^ successfully replaced Zn^2+^ at some points in the lattice [16,17,26]. Additionally, Mg peak intensities increased concurrently with increasing dopant amounts. Overall, XPS measurements indicate that the energy level of ZnO was slightly shifted towards a higher binding energy when Mg was incorporated, likely resulting from Zn-O-Mg bond formation when Mg substituted Zn [26]. These changes in chemical and energy states could influence the charge-transfer process between the substrate and molecules to enhance Raman signals.

Changes in the optical properties of ZnO when doped with Mg were investigated by UV-vis absorption spectroscopy. As observed in Figure 3a, the absorption edges of the Mg-doped ZnO nanoparticles are red-shifted compared to pure ZnO nanoparticles, and an increase in the red-shift was observed with an increased dopant amount. Wavelength values of 352, 355, 361, 363, and 365 nm were obtained for 0%, 2%, 5%, 7%, and 10% doping, respectively. The observed shift in absorption with the doping concentration could be due to factors such as particle size, oxygen deficiency, and lattice strain resulting from Mg doping [25]. The optical band gap was estimated from Tauc’s relationship, (αhv)^1/n^ = A(hv − E_g_), where α = absorption coefficient, h = Plank’s constant, v = frequency of incident photon, E_g_ = energy gap, A = proportionality constant, and n = transition constant [21]. As shown in Figure 3b, when the plots of (αhv)^2^ vs. (hv) were made for pure and doped ZnO nanoparticles, and taking n as 1/2 since the materials allow for direct transition, the linear portion can be extrapolated to obtain the band gap [19,25]. The band gaps of the Mg-doped ZnO nanoparticles were determined to be lower compared to those of pure ZnO nanoparticles, with values of 3.36, 3.28, 3.17, 3.08, and 3.01 eV obtained for 0%, 2%, 5%, 7%, and 10% doping, respectively. Decreases in the band gap are thought to be due to strong quantum confinements and enhancement in the substrate’s surface-area-to-volume ratio [25]. Together with the XPS data, the increase in the red-shift and the decrease in bandgap energy indicate the presence of Mg^2+^ in the Zn^2+^ site of the ZnO lattice [25,26,27]. Consequently, the narrowed band gap could cause the energy of the incident light to be sufficient for a charge-transfer transition between the substrate and the molecule. These data show tunability with respect to the bandgap values that can then in turn be used to select the appropriate synthesis method based on desired energy levels. 

### 3.2. Raman Signal Enhancement

DTNB was employed as a model compound due to its excellent chemistry to determine the enhancement efficiency of the synthesized colloidal substrates in SERS studies. Thus, the Raman spectra of DTNB colloidally dispersed with the substrates were acquired. As shown in Figure 4a, peaks of DTNB were observed at 850 cm^−1^, 1058 cm^−1^, 1150 cm^−1^, 1341 cm^−1^, and 1557 cm^−1^, corresponding to nitro scissoring vibration, C–N stretching and C–N bending modes, CH_3_ rocking, symmetric nitro stretch vs. (NO_2_), and an aromatic ring stretching mode, respectively [33]. As mentioned, the chemical enhancement effect is often characterized by interactions between the VB, CB, HOMO, and LUMO of the semiconductor substrates and molecule/target of interest [9]. For instance, as shown in Figure 4b, the CB and VB of the ZnO substrate are −1.9 and −5.2 eV [19,20], while the HOMO and LUMO of DTNB are −7.35 and −3.64 eV, respectively [34]. The excitation energies required for excitonic transition and molecular transition are 3.3 and 3.71 eV, respectively, whereas the energy provided by the 785 nm laser is 1.58 eV, which is not enough for excitonic and molecular transition to take place. In addition, the charge-transfer transition from the HOMO of DTNB to the CB of ZnO will require 5.45 eV, which is also not possible. On the other hand, the energy needed for the charge-transfer transition from the VB of ZnO to the LUMO of DTNB is 1.56 eV, which can be provided by the laser. The VB-LUMO transition path seems to be the dominant route for charge transfer in this study, which is also consistent with the work of Kim et al. [10]. This provides the possibility of selecting molecules based on their energy level compatibility with the substrate, which could also play a significant role in developing a SERS substrate or nanotag for future bioanalytical studies.

When ZnO was doped with Mg, a more enhanced signal was observed with DTNB dispersed in 2% Mg-doped-ZnO compared with the other doping concentrations (See Table A1). Previously, a maximum signal enhancement was reported at low doping percentages [17,20], and it was suggested that at higher doping concentrations, higher defect concentrations could cause electron–hole recombination and thus compete with the charge transfer from the substrates to the molecules. This indicates that the band gap does not necessarily have a direct relationship with signal enhancement but that appropriate synthesis conditions can nevertheless be selected to influence electronic compatibility with a target molecule or Raman reporter molecule (in the case of SERS nanotags). In our work, using the 1341 cm^−1^ peak, the average Raman intensity of DTNB dispersed in the 2% Mg-doped-ZnO substrate is about three times higher than the Raman intensity of DTNB, while the Raman intensity of DTNB dispersed in pure ZnO substrates was less than twice the Raman intensity of DTNB. This indicates that the enhancement of DTNB by ZnO colloidal substrates can be tailored by appropriate doping. When Mg^2+^ substitutes Zn^2+^, there is an increase in the number of oxygen vacancies and electron concentration due to differences in the radius and electronegativity between Zn^2+^ and Mg^2+^ [16]. These surface defects introduce a new energy level below the conduction band, which can act as electron traps to reduce photoexcitation charge recombination and serve as an intermediate state (see Figure A2) for electron transfer between the semiconductor substrate and target molecule [16,17,20]. Besides the new surface-state energy level, the impact of doping-induced bandgap shrinkage on the charge transfer between semiconductors and molecules has also been reported [19]. In this case, the Mg dopant caused a red-shift in the absorption wavelength, reducing the band gap and thus promoting charge transfer between ZnO and DTNB. In other words, the required energy for charge transfer is further reduced, which allows for an easier inter-band charge transition between the VB, surface-state level of ZnO, and the LUMO of DTNB and hence produces a stronger Raman signal. Generally, doping can change the physical characteristics and band gap of ZnO, as confirmed by FE-SEM and UV-vis. These could result in signal enhancement from a size-dependent effect or bandgap shrinkage [19]. In our work, the size-dependent effect is less likely because, despite the slightly larger size of 5–10% doped samples compared to pure ZnO, we did not observe a higher Raman signal. This observation implies that, in this work, the particle size could have directly impacted the bandgap but not necessarily the signal enhancement. In addition, it has been reported that optimal Raman signals resulting from size-dependent charge-transfer resonance can be observed in ZnO with a 28 nm particle size [35], which is smaller than our samples. Therefore, the bandgap shrinkage rather than the size effect is a more probable cause of the signal enhancement observed.

### 3.3. Stability Study

Nanoparticle stability plays a critical role in their use in various applications, including biological and environmental sensing. In a recent study, Cao et al. determined the stability of ZnO nanoparticles suspended in ethanol by plotting the absorbance of the nanoparticles at 370 nm against time for 24 h [36]. They noticed an excellent suspension performance when their synthesis conditions were optimized. However, biological environments contain biomacromolecules, such as electrolytes, proteins, and lipids [28,37], which subject nanoparticles to intermolecular and surface forces (e.g., van der Waals, the repulsive electrostatic double layer, and structural forces) [28,30]. Due to this complexity, simple aqueous solutions cannot adequately evaluate the interaction of nanoparticles in complex biological systems [28]. To test the substrates’ stability in a biological milieu, UV-vis was used to measure changes in the absorption wavelength (352–365 nm) over 24 h in standard cell culture media, supplemented with 1% penicillin/streptomycin and 10% fetal bovine serum. A decrease in absorbance with time is indicative of particles aggregating and settling down, mimicking how substrates may behave in a biological environment exposed to proteins, biological pH, and salts. As shown in Figure 5 (inset), in 60 min in our study, all substrate samples displayed high stability, as indicated by the horizontal slope in the graph. As time progressed, a moderate change in the slope, as illustrated by the pure ZnO and 2% Mg-doped ZnO substrates, indicated a more stable substrate, while at a higher doping percentage, it can be observed that the slope has a higher gradient, indicating some aggregation. These could possibly have resulted from differences in the surface energy of the doped substrates, resulting in the agglomeration of smaller particles within the preparations or their interactions with proteins present in the sample. In addition, variations in particle size could have reduced stability past the one-hour time point. The aggregation of nanoparticles is system-dependent, and various factors could influence the stability of nanoparticles in biological media. Such factors include the physical and chemical compositions of the nanoparticles, surface coatings, media properties, and the presence of macromolecules. Generally, the stability of substrates can be improved with surface modification by electrostatic, steric, or electrosteric stabilization [28].

### 3.4. Cytotoxicity Study

To study the effect of the synthesized substrates on cytotoxicity, MCF-7 cells were exposed to pure ZnO, 2% Mg-doped ZnO, 5% Mg-doped ZnO, 7% Mg-doped ZnO, and 10% Mg-doped ZnO substrates at a concentration range of 3 to 100 μg/mL for 24 h and evaluated by the MTT assay. As shown in Figure 6a, the synthesized substrates did not show a significant impact on cell viability at a concentration range of 3 to 25 μg/mL. However, at higher concentrations (50 and 100 μg/mL), the cell viability significantly decreased by about 50%. This decrease in cell viability at the 50 and 100 μg/mL doses was statistically significant with a *p*-value < 0.05. The cytotoxicity of these substrates was further assessed at different time intervals (24 h, 36 h, and 48 h). As shown in Figure 6b, the cytotoxicity of the 25 μg/mL concentration of the substrates was relatively low, with no statistically significant difference at these time intervals. In addition, at concentrations lower than 25 μg/mL (i.e., 3 to 12 μg/mL), the substrates showed extremely low cytotoxic behavior, with little or no difference between the time intervals (see Figure A3). On the other hand, when the cells were treated with 100 μg/mL (Figure 6c), cell viability decreased by approximately two-fold compared to the control at all time intervals.

When comparing substrates, it is worth noting that no significant differences in cytotoxicity were observed when administered at a low dosage and the same time interval. In other words, the process of doping itself did not induce significant changes in cytotoxicity. However, the concentration of the substrate played a major role, as cell viability was greatly impacted at high doses (50 and 100 μg/mL) in all incubation periods. Cierech et al. [38] previously reported that there was no significant difference in the viability of Hela cells treated with 1–20 μg/mL and the control group after 24 h incubation, while they observed a significant decrease in cell viability at concentrations of 30, 50, and 100 μg/mL. Hamidian et al. [39] also reported that low concentrations of undoped ZnO had low toxic effects on brain glioblastoma cells; however, with increasing concentrations (50 and 500 μg/mL) of ZnO, the cell viability greatly decreased. This indicates that cytotoxicity can be avoided at low concentrations (<25 μg/mL) of these substrates, which can be expected for their use in SERS bio-detection/bio-imaging applications.

## 4. Conclusions

This study shows the potential for developing colloidal semiconductor-based substrates with application in SERS via substitutional doping. Pure and Mg-doped ZnO were synthesized using a co-precipitation method, and SEM and XPS analyses confirmed the presence of Zn and O in the pure sample and the successful incorporation of Mg into doped samples. UV-vis showed a red-shift in wavelength with increasing dopant concentration, while the calculated band gap decreased (3.3–3.01 eV) with increasing dopant content. The maximum SERS enhancement was obtained at a 2% doping concentration. All substrates showed high stability in the first hour of dissolution in a cell media; however, pure ZnO and 2% Mg-doped ZnO showed the best stability over 24 h. Finally, it was observed that there was no significant cytotoxic effect resulting from doping Mg with ZnO. This study shows that by applying an appropriate amount of the dopant, ZnO can be tailored to impart changes in enhancement efficiency. It is expected that this work could serve as a platform for SERS-based molecular sensing and bio-detection.

## Figures and Tables

**Figure 1 nanomaterials-12-03564-f001:**
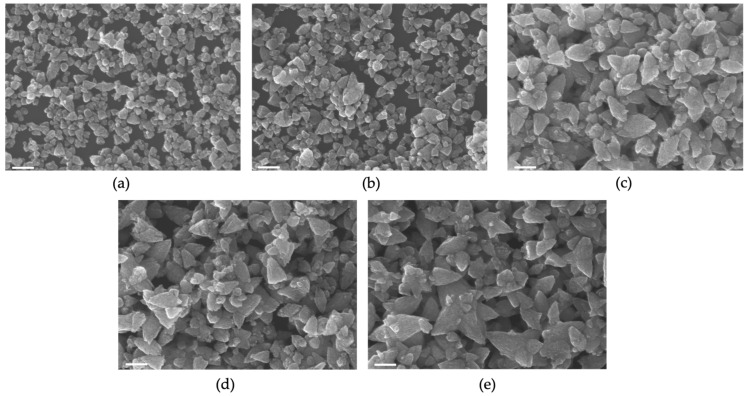
FE-SEM images (scale bar: 200 nm) of (**a**) pure ZnO nanoparticles; (**b**) 2% Mg-doped nanoparticles; (**c**) 5% Mg-doped nanoparticles; (**d**) 7% Mg-doped nanoparticles; (**e**) 10% Mg-doped nanoparticles.

**Figure 2 nanomaterials-12-03564-f002:**
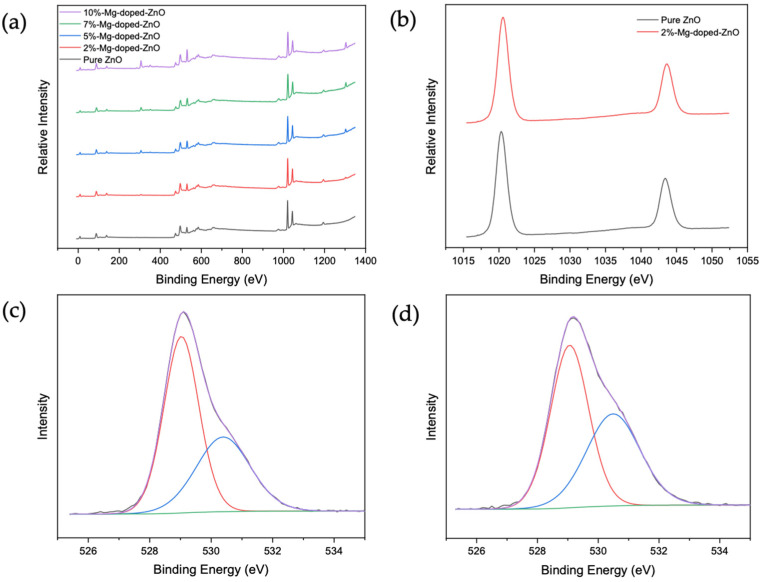
(**a**) XPS full scan of pure and Mg-doped ZnO nanoparticles; (**b**) high-resolution Zn scan of pure ZnO and 2% Mg-doped ZnO nanoparticles; (**c**) high-resolution O1s scan of pure ZnO nanoparticles; (**d**) high-resolution O1s scan of 2% Mg-doped ZnO nanoparticles.

**Figure 3 nanomaterials-12-03564-f003:**
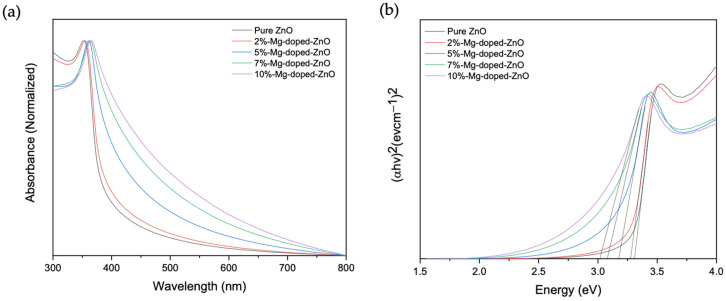
(**a**) Normalized absorbance spectra of pure and ZnO nanostructures; (**b**) Tauc’s plot showing bandgap values, which were obtained by extrapolating the linear portions of the plot and are depicted as black lines.

**Figure 4 nanomaterials-12-03564-f004:**
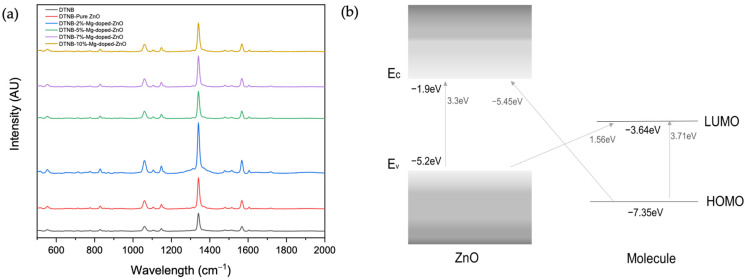
(**a**) Raman spectra of 10^−2^ M DTNB enhanced by ZnO and Mg-doped ZnO SERS substrates; (**b**) schematic diagram of energy band alignment between ZnO and DTNB.

**Figure 5 nanomaterials-12-03564-f005:**
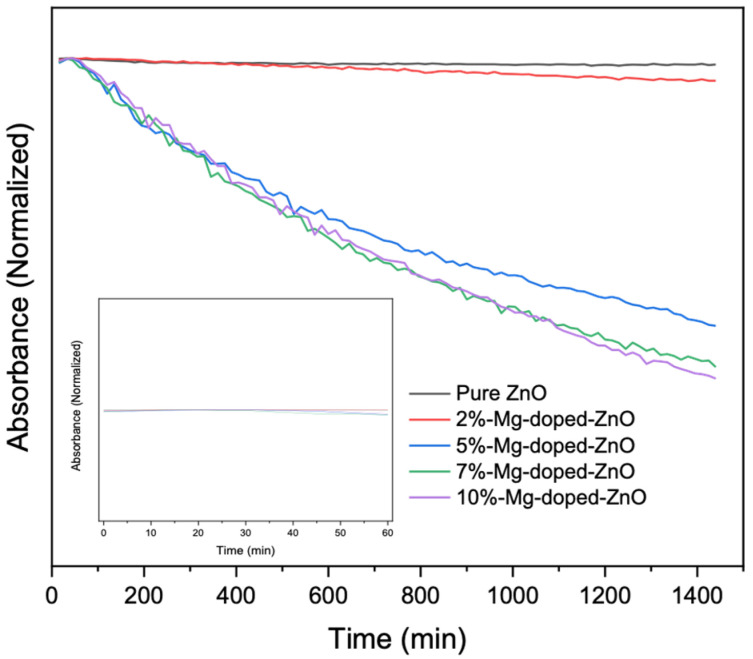
UV-vis absorbance plot comparing the stability of pure and Mg-doped ZnO nanoparticles. Inset: represents stability in the first 60 min.

**Figure 6 nanomaterials-12-03564-f006:**
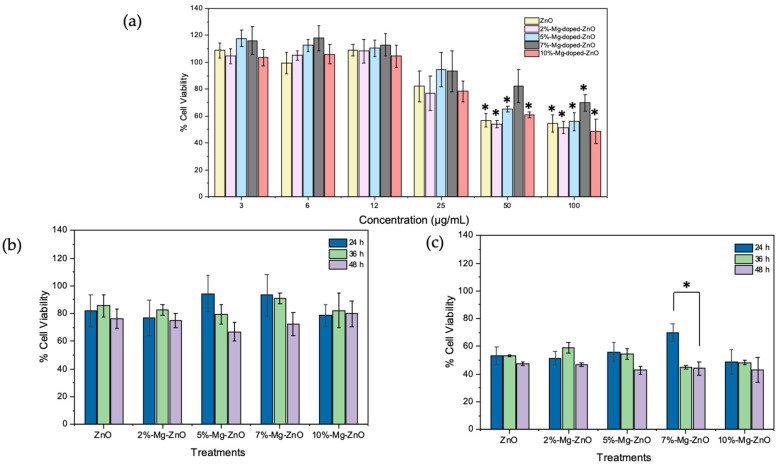
(**a**) Effect of ZnO nanoparticles and Mg-doped ZnO nanoparticles on the viability of MCF-7 cells after 24 h, as determined by MTT assay. It was found that high doses, e.g., 50 and 100 μg/mL, show significant cytotoxicity; (**b**) viability of MCF-7 cells treated with 25 μg/mL ZnO nanoparticles and Mg-doped ZnO nanoparticles for 24 h, 36 h, and 48 h; (**c**) viability of MCF-7 cells treated with 100 μg/mL ZnO nanoparticles and Mg-doped ZnO nanoparticles. Cell viability was measured after 24 h, 36 h, and 48 h by MTT assay. Data are represented as the mean of triplicate ± SEM (* *p* < 0.05).

## Data Availability

Data are available within this article.

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
