# Peer review of "Mg-Doped ZnO Nanoparticles with Tunable Band Gaps for Surface-Enhanced Raman Scattering (SERS)-Based Sensing"

_nanomaterials, 2022, doi:10.3390/nano12203564_

Round 1

Reviewer 1 Report

The manuscript by Samuel Adesoye et al. demonstrates an Mg-doped ZnO that can change the bandgap by manipulating the Mg doping amount for SERS substrates to achieve enhanced signal intensity. The idea is good, but the results don't fully support the idea. At the same time, reports of improving SERS detection by doping can be seen everywhere (10.1039/C8TC06588H, 10.1007/s10854-019-02416-4). Also, compared with other reports, the improvement in this paper is not significant. Therefore, after carefully evaluating this manuscript, I do not find the manuscript suitable for publication in Nanomaterials because of the lack of clear evidence for novelty and scientific advancement compared to the state of the art. Detailed comments are below:

1. The abbreviation ‘MTT’ on line 19 on the first page is not indicated

2. FE-SEM cannot obtain the hexagonal crystal-shaped structure in Figure 1

3. The band information about DTNB should be supplemented by some test proofs, not just relying on citations.

4. ‘When ZnO was doped with Mg, a more enhanced signal was observed with DTNB dispersed in 2% Mg-doped-ZnO compared with the other doping concentrations.’ Should the data be made more convincing by adding ordinates etc.? In addition, as the core point of this paper, its enhanced capability should be further demonstrated.

5. Please explain in detail the reasons for the considerable difference in the stability of different concentrations in Figure 5 and the sudden decrease in stability after the first hour at a higher doping percentage.

6. Pay attention to the number of conclusions.

Reviewer 2 Report

The study shows the potential for developing new semiconductor-based substrates with application in SERS via substitutional doping. ZnO nanoparticles were synthesized by co-precipitation and doped with Mg at concentrations ranging from 2-10%. The absorption edge of Mg-doped ZnO nanoparticles was red-shifted compared to pure ZnO nanoparticles. And the band gap decreased with increasing Mg doping, while the highest Raman enhancement was observed at a 2% doping ratio. No significant cytotoxic effects were observed at low concentrations (i.e., 3 to 12 μg/mL). The phenomenon is interesting, which provide evidence for the tunability of ZnO substrates and may serve as a platform for applications in molecular biosensing. However, the interpretation is not well discussed at the moment, some questions should be addressed:

1. The band gap shift resulting from doping ZnO with Mg has been studied in different experiments exploring various applications. What the main breakthrough of the manuscript comparing with these studies?

2.The author stated that the decreases in the band gap are thought to be due to strong quantum confinements and enhancement in the substrate’s surface area to volume ratio. In the following text (line 291), the author pointed out that the band-gap shrinkage rather than size effect is a more probable cause of the signal enhancement observed. But the size of sample will affect the substrate’s surface area to volume ratio. The author needs to give an explanation.

3. With the increase of doping concentration, the absorption spectrum is red-shifted, indicating that the band gap decreases with the increase of concentration, but the Raman enhancement is not the strongest at the concentration of 10%, and the authors need to explain.

4.When doped with Mg, a defect energy level was introduced, which can act as electron traps to reduce photoexcitation charge recombination and serve as an intermediate state for electron transfer between the semiconductor substrate and target molecule. The author should provide a schematic diagram of the energy after doping of Mg, obtain the position of the defect energy level from the XPS data, and draw the defect energy level.

5.The picture size needs to be modified

Reviewer 3 Report

The manuscript entitled “Mg-Doped ZnO Nanoparticles with Tunable Band Gaps for Surface-Enhanced Raman Scattering (SERS)-Based Sensing” describes Mg-Doped ZnO Nanoparticles with Tunable Band Gaps. The essential characterization methods such as FTIR, XRD, HR(TEM) and sensing measurements were not performed. Thus, some major issues are needed to be addressed first before the recommendation of this work for publication to Nanomaterials.

Comments:

  1.   The author should provide some quantitative information in the abstract section.
  2. The author should study important characterization such as FTIR for the confirmation of Mg-doped ZnO nanoparticles, XRD for the confirmation of crystallinity, (HR)TEM for the confirmation of the average diameter of a small range of ZnO nanoparticles.
  3. The author stated in the abstract section, Morphology and size distribution were obtained by scanning electron microscopy (SEM). But there is no size distribution measurement in the results and discussion section. The author should carry out this.
  4.  The author should cite some references in section 3.4. Cytotoxicity Study.

Round 2

Reviewer 1 Report

The authors addressed properly the comments and the paper can be considered for publication.

Reviewer 2 Report

All the comments have been addressed in this revised manuscript